# Advancing Treatment Outcomes for Peritoneal Surface Malignancies in Low- and Middle-Income Countries: Insights from the First Multicenter Study in North Africa

**DOI:** 10.3390/cancers17132113

**Published:** 2025-06-24

**Authors:** Amine Souadka, Hajar Habbat, Amin Makni, Mourad Abid, Zakaria El Mouatassim, Amin Daghfous, Zakia Korjani, Wael Rebai, Mouna Ayadi, Wafa Hania Messai, Mohammed Anass Majbar, Amine Benkabbou, Raouf Mohsine, Abdelilah Souadka

**Affiliations:** 1Surgical Oncology Department, National Institute of Oncology, Mohammed V University, Rabat 10000, Morocco; hajarhabbat1@gmail.com (H.H.); zakaria.elmouatassim@um5s.net.ma (Z.E.M.); a.majbar@um5r.ac.ma (M.A.M.); a.benkabbou@um5r.ac.ma (A.B.); raoufmohsine@gmail.com (R.M.); abdelilah.souadka@gmail.com (A.S.); 2Surgical Department A, Rabta Hospital, Tunis 1007, Tunisia; aminmakni1@gmail.com (A.M.); aminedaghfous@yahoo.fr (A.D.); rebaiwael@gmail.com (W.R.); 3Surgical Oncology Department, Batna Cancer Institute, Batna, Algeria; mo.abid@univ-alger.dz (M.A.); zakikor16@gmail.com (Z.K.); messaihania@gmail.com (W.H.M.); 4Medical Oncology Department, Salah-Azaiz Institute, Tunis 1006, Tunisia; mounaayadi28@gmail.com; 5Surgical Oncology Department, Private Oncology Center, Rabat, Morocco

**Keywords:** peritoneal surface malignancies (PSM), cytoreductive surgery (CRS), hyperthermic intraperitoneal chemotherapy (HIPEC), low- and middle-income countries (LMIC)

## Abstract

Cytoreductive surgery (CRS) with or without hyperthermic intraperitoneal chemotherapy (HIPEC) is a key treatment for peritoneal surface malignancies (PSM), yet its implementation in low- and middle-income countries (LMICs) remains challenging due to resource constraints. This multicenter study from North Africa demonstrates the feasibility and efficacy of CRS+HIPEC in a resource-limited setting, achieving survival outcomes comparable to high-income countries. The findings highlight the importance of structured training, optimized HIPEC protocols, and multidisciplinary collaboration in expanding PSM programs in LMICs. Future efforts should focus on improving patient selection, enhancing perioperative care, and ensuring long-term program sustainability.

## 1. Background

Peritoneal surface malignancies (PSM) represent a heterogeneous group of cancers that share a tendency to disseminate within the peritoneal cavity. These malignancies may either arise as primary tumors, such as peritoneal mesothelioma or as secondary metastases from other cancers [1]. Traditionally, peritoneal carcinomatosis was considered a terminal condition with limited treatment options. However, the introduction of cytoreductive surgery (CRS) combined with hyperthermic intraperitoneal chemotherapy (HIPEC) has revolutionized the treatment landscape. This approach offers certain patients the possibility of prolonged survival and, in some cases, even a potential cure. Consequently, PSM is increasingly viewed not just as an end-stage disease but as a loco-regional condition that, with appropriate management, can be treated with curative intent. As expertise with CRS and HIPEC expands, these treatments are expected to become more accessible, improving outcomes for a broader range of patients.

Nevertheless, significant regional disparities persist, particularly in low- and middle-income countries (LMICs), where access to specialized therapies remains limited. Recent studies suggest that, despite such challenges, HIPEC can be effectively integrated into treatment protocols in resource-constrained settings [2,3,4,5,6,7].

In the setting of North Africa, which is experiencing a rising burden of cancer, research on the management of PSM remains sparse [4].

To address this gap, the present study investigates the treatment outcomes of PSM in Morocco, Tunisia, and Algeria. It examines patient characteristics, treatment strategies, and survival outcomes, with a focus on the implementation of HIPEC and its impact on prognosis. By evaluating the feasibility and effectiveness of these interventions in resource-limited settings, this study aims to contribute valuable insights. Furthermore, it seeks to guide the development of sustainable PSM treatment programs in LMICs, facilitating the adaptation of CRS and HIPEC to healthcare systems with limited resources while striving to achieve survival outcomes comparable to those seen in high-income countries.

## 2. Methods

### 2.1. Study Design

This retrospective study analyzed data from patients with peritoneal surface malignancies (PSM) treated with cytoreductive surgery (CRS), treated with curative intent with or without hyperthermic intraperitoneal chemotherapy (HIPEC), across oncology centers in Rabat (Morocco), Tunis (Tunisia), and Batna (Algeria) between January 2014 and December 2020. These centers were selected for their ability to perform advanced treatments like CRS and HIPEC in the region, as well as their close proximity, similar clinical protocols, and shared practice settings. As such, they provide a representative sample of healthcare in North Africa and reflect the challenges and opportunities of managing PSM in low- and middle-income countries (LMICs) (Figure 1).

### 2.2. Patient Selection

Patients were included if they had a confirmed diagnosis of PSM originating from colorectal cancer (CRC), pseudomyxoma peritonei (PMP), gastric cancer (GC), ovarian cancer (OC), or mesothelioma and had undergone CRS, with or without HIPEC. Complete medical records, including follow-up data, were required for inclusion. Patients with incomplete medical records or missing follow-up data were excluded.

Systemic neoadjuvant chemotherapy was selectively administered, depending on tumor type and institutional MDT recommendations: For colorectal cancer, neoadjuvant FOLFOX or FOLFIRI was usually given to patients with high PCI or suspected nodal involvement. For gastric cancer, perioperative chemotherapy (based on the ECX or FLOT regimen) was used. For ovarian cancer, when needed, patients received neoadjuvant carboplatin-paclitaxel chemotherapy prior to interval CRS. For pseudomyxoma peritonei and mesothelioma, no systemic neoadjuvant therapy was administered.

### 2.3. Data Collection

Data were extracted from prospective institutional databases, systematically tracking patient information from diagnosis through follow-up. Variables collected included demographic details (age, sex), clinical parameters (ASA score, ECOG performance status, use of neoadjuvant therapy), tumor-specific factors (primary tumor site, Peritoneal Cancer Index [PCI]), and perioperative details (Completeness of Cytoreduction [CC] score, administration of HIPEC, including regimen and duration). Outcome measures included overall survival (OS), disease-free survival (DFS), occurrence of complications, and classification of complications according to the Clavien–Dindo system [8].

### 2.4. Data Standardization

To ensure consistency, data from all participating centers were consolidated into a single dataset, with uniform variable definitions and formatting applied across institutions. PCI scores were categorized into four groups (1–6, 7–12, 13–19, >19). Cytoreduction status was classified as CC-0/CC-1 (complete or minimal residual disease) and CC-2 (incomplete cytoreduction). Severe complications were defined as Clavien–Dindo grade > IIIa [5]. OS was measured from the date of surgery to death or last follow-up, while DFS was defined as the time from surgery to disease recurrence or last follow-up.

### 2.5. Statistical Analysis

Descriptive statistics were used to summarize patient characteristics. Continuous variables were presented as medians with interquartile ranges (IQR), and categorical variables as counts and percentages. The association between clinical variables and severe complications was assessed using the Mann–Whitney U test for continuous variables and Fisher’s exact test for categorical variables. Predictive factors for severe complications were evaluated using univariate and multivariate logistic regression analyses.

Survival analysis was conducted using the Kaplan–Meier method, with comparisons between groups performed using the log-rank test. Cox proportional hazards regression was applied to determine the impact of independent variables on OS and DFS. Variables significant in univariate analysis were included in the multivariate model to identify independent prognostic factors.

All statistical analyses were performed using SPSS Statistics for Windows, version 24.0 (IBM Corp., Armonk, NY, USA). A two-tailed *p*-value of <0.05 was considered statistically significant.

### 2.6. Ethical Considerations

This study follows the STROBE (Strengthening the Reporting of Observational Studies in Epidemiology) guidelines for observational research [9]. Ethical approval was obtained from the institutional review board (CIC—Centre des Investigations Cliniques) at the University Hospital Ibn Sina, in accordance with Moroccan law (Law 28/13—art.2). The study also received approval from the local committee of the National Institute of Oncology in Rabat, INO/13/24, as well as from local committees in participating centers in Batna and Tunis. All procedures involving human subjects adhered to the ethical standards of institutional and national research committees, as well as the 1964 Helsinki Declaration and its subsequent amendments or equivalent ethical guidelines.

Participants were fully informed about the study’s objectives at the time of the procedure and voluntarily provided their informed consent to take part in the research.

## 3. Program Details

### 3.1. Preoperative Evaluation

All patients received routine preoperative lab work-ups. Tumor marker levels were additionally measured in all patients based on the malignancy type: carcinoembryonic antigen (CEA) and CA19-9 for colorectal cancer (CRC), gastric cancer (GC), and pseudomyxoma peritonei (PMP); and CA125 for ovarian cancer (OC). Imaging protocols included routine contrast-enhanced computed tomography (CT) scans for all patients, with magnetic resonance imaging (MRI) performed selectively based on clinical indications.

All cases were reviewed in multidisciplinary team (MDT) meetings, which consisted of radiologists, medical oncologists, surgical oncologists, and radiation oncologists, in order to assess preoperative imaging findings, determine the extent of disease, and define the treatment strategy.

### 3.2. Technique

#### 3.2.1. Anesthesia

All patients undergoing CRS with or without HIPEC received general anesthesia administered by an experienced anesthesiology team. Preoperative assessments included evaluations of cardiovascular and respiratory function, with tailored optimization strategies to minimize perioperative risk. Anesthetic protocols prioritized hemodynamic stability and temperature regulation during the procedure, with continuous intraoperative monitoring of vital parameters, including core body temperature, central venous pressure, and arterial blood gases. Advanced airway management and mechanical ventilation techniques were employed to ensure adequate oxygenation and ventilation throughout the extended surgical duration. Postoperative analgesia was managed using multimodal approaches.

#### 3.2.2. Cytoreductive Surgery

Cytoreductive surgery and peritonectomy were conducted for curative intent in accordance with standardized guidelines tailored to the specific type of malignancy and the extent of peritoneal carcinomatosis [10].

The extent of the disease was quantified using the Peritoneal Cancer Index (PCI), which evaluates lesion size (scored 0–3) across 13 abdominal regions, yielding a total score ranging from 1 to 39 [11]. For analysis, PCI scores were classified by tumor origin and stratified into four categories: 1–6, 7–12, 13–19, and >19.

The surgical procedures involved the resection of up to five anatomical regions, including the right and left diaphragmatic peritoneum, right and left upper abdominal quadrants, anterior parietal peritoneum, pelvic peritoneum, and infragastric omentum. Regions not subjected to resection were treated using electrocautery to achieve coagulation.

In cases of ovarian carcinomatosis, pelvic and aortic lymphadenectomy was routinely performed alongside cytoreductive surgery (CRS). When coloproctectomy was required, the procedure ensured a minimum residual small bowel length of 2 m, while total gastrectomy was contraindicated. For gastric carcinomatosis, partial gastrectomy was performed in combination with CRS, followed by digestive tract reconstruction using a Roux-en-Y anastomosis. Similarly, for rectal cancer with carcinomatosis, partial or total mesorectal excision was carried out as part of CRS.

Patients who underwent unplanned splenectomy or spleno-pancreatectomy were administered antibiotics and vaccinations within one month postoperatively, provided no complications were observed.

Completeness of cytoreduction (CC) was assessed using Sugarbaker’s classification, defined as follows: CC0 for no visible residual disease, CC1 for residual nodules <2.5 mm, CC2 for residual nodules measuring 2.5 mm–2.5 cm, and CC3 for residual nodules exceeding 2.5 cm [10].

#### 3.2.3. Hyperthermic Intraperitoneal Chemotherapy (HIPEC)

HIPEC procedures at the National Institute of Oncology in Rabat were performed using the closed continuous circulation system technique using the ThermoChem HT-2000^®^ (ThermaSolutions, 1889 Buerkle Road White Bear Lake, MN, USA, https://www.thermasolutions.com/) operator, whereas, in the other centers, they were performed using the GAMIDA SunChip 1^®^ (https://gamida.fr/chip-chit/, accessed on 13 June 2025) maintaining the chemotherapy solution at a target temperature of 42 °C.

For all tumor types except ovarian cancer, the HIPEC regimen consisted of oxaliplatin (460 mg/m^2^) administered over 30 min, preceded by systemic chemotherapy with fluorouracil (400 mg/m^2^) and folinic acid (20 mg/m^2^) delivered intravenously 20 min before the procedure. For ovarian cancer, a fractionated cisplatin regimen was employed to mitigate nephrotoxicity, given the unavailability of sodium thiosulfate at our center.

#### 3.2.4. Postoperative Management and Identification of Complications

Following CRS and HIPEC procedures, patients were closely monitored in the intensive care unit to ensure early identification and management of potential complications. Standard postoperative care included continuous monitoring of vital signs, such as heart rate, blood pressure, respiratory rate, oxygen saturation, and body temperature, alongside regular clinical assessments. Early mobilization and kinesiotherapy were encouraged to minimize risks associated with immobility, such as venous thromboembolism.

Routine laboratory tests, including complete blood counts and electrolyte panels, were performed to detect abnormalities indicative of complications such as infections, electrolyte imbalances, or renal impairment. Imaging studies, such as abdominal ultrasonography or computed tomography (CT) scans, were obtained as needed to investigate suspected postoperative issues like fluid collections or anastomotic leaks.

Follow-up assessments and postoperative care were performed according to the Enhanced Recovery After Surgery (ERAS) protocol, which emphasized multidisciplinary collaboration to ensure the timely management of complications and optimize recovery [12]. Patients who developed complications requiring reoperation were managed according to established institutional protocols to mitigate morbidity and improve outcomes.

### 3.3. Center-Specific Protocols and Practices in the PSM Program

The PSM programs across Morocco, Tunisia, and Algeria vary in terms of several aspects, such as program initiation, available resources, surgical training, and tumor-specific protocols.

In Morocco, the PSM program officially started in 2014 and is currently led by a single surgeon performing these procedures [5]. The surgeon completed a two-year specialized fellowship at the Institut Gustave Roussy (IGR) in France. On-site mentorship was not available. However, the team benefits from ongoing remote guidance from national experts in multi-organ oncological surgery. The surgical learning curve was reached earlier than that of the anesthesiology team, leading to initial discrepancies in outcomes that were resolved through improvements in communication, research collaboration, and team trust. The lead surgeon is actually mentoring new team members and overseeing training initiatives.

In Tunisia, the PSM program began in 2003 with an artisanal HIPEC device and transitioned to using the SunChip device in 2009. The team currently includes ten surgeons, two of whom completed specialized training abroad, including six months at the Institut Gustave Roussy (IGR) in France. These surgeons mentor new team members and oversee training initiatives. Surgical proficiency for cytoreductive surgery was achieved after 10 cases.

In Algeria, the PSM program was established in 2015 with two surgeons trained through mentorship with Pr. Marc Pocard and short-term placements in specialized centers. On-site mentorship and training are provided to new teams. The surgical learning curve for cytoreductive surgery requires approximately 20 cases, while HIPEC procedures are simpler but rely heavily on collaboration with anesthesiologists.

All participating centers follow standardized French treatment protocols, which include the following:Patient selection guided by multidisciplinary tumor board (MDT) discussions, incorporating imaging, Peritoneal Cancer Index (PCI) scoring, and ECOG/ASA evaluations.Cytoreductive surgery (CRS) performed using Sugarbaker’s peritonectomy techniques, with resections tailored to PCI findings.HIPEC protocols consisting of a short-duration oxaliplatin-based regimen (460 mg/m^2^ over 30 min) for colorectal cancer, pseudomyxoma peritonei, and gastric cancer, and a cisplatin-based regimen for ovarian cancer.Postoperative management following Enhanced Recovery After Surgery (ERAS) guidelines, including early mobilization, thromboprophylaxis, infection prevention, and nutritional support.

## 4. Results

Out of 954 patients presented at the multidisciplinary team (MDT) meetings to discuss resectability, 391 patients were selected for inclusion in this study. A detailed overview of the study population, including demographic and clinical characteristics, is outlined in Table 1.

### 4.1. Tumor Characteristics

Colorectal cancer (CRC) represented the most common primary tumor (43.0%, n = 168), followed by pseudomyxoma peritonei (PMP) (36.8%, n = 144), ovarian cancer (12.0%, n = 47), gastric cancer (6.4%, n = 25), and mesothelioma (1.8%, n = 7).

### 4.2. Treatment Characteristics

Complete cytoreduction (CC-0/1) was achieved in 88% of cases, and HIPEC was performed in 39%. The most common surgical procedures included peritonectomy involving more than four regions (59.1%, n = 231), proctectomy (27.7%, n = 108), spleno-pancreatectomy (12.8%, n = 50), and bowel anastomosis involving more than two sites (15.3%, n = 60).

### 4.3. Postoperative Outcomes

Postoperative complications were observed in 42.2% (n = 165) of patients, with severe complications (Clavien–Dindo grade ≥ III) occurring in 22.3% (n = 87). The mortality rate (Clavien–Dindo grade V) was 5.6% (n = 22). The most common complications were postoperative peritonitis (7.4%), pulmonary complications (5.6%), sepsis (4.9%), deep abscess (4.6%), evisceration, abdominal fistula, and anastomotic leak (3.6%).

Multivariate logistic regression identified several independent predictors of severe complications. An ASA score ≥2 was associated with a significantly increased risk (OR = 2.228, 95% CI: 1.309–3.793; *p* = 0.003). Spleno-pancreatectomy emerged as a strong predictor (OR = 3.803, 95% CI: 1.765–8.197; *p* < 0.001), along with incomplete cytoreduction (CC-2; OR = 3.830, 95% CI: 1.765–8.297; *p* < 0.001) and HIPEC (OR = 2.364, 95% CI: 1.369–4.080; *p* = 0.002). Detailed results are summarized in Table 2.

### 4.4. Survival Outcomes

The median overall survival (OS) for the cohort was 68 months, with 1-year, 5-year, and 7-year survival rates of 97%, 56%, and 37%, respectively (Figure 2).

A total of 146 deaths (37.3%) were recorded for OS events, distributed as follows: colorectal cancer (CRC), 72; pseudomyxoma peritonei (PMP), 34; ovarian cancer (OC), 22; gastric cancer (GC), 16; and mesothelioma, 2. For DFS, a total of 173 events (recurrence or death; 44.2%) were reported: CRC, 84; PMP, 39; OC, 26; GC, 21; and mesothelioma, 3.

Patients with PCI 1–6 demonstrated the highest median OS (79 months), while those with PCI > 19 had the lowest (60 months). The use of HIPEC was associated with improved survival (HR = 0.634, 95% CI: 0.434–0.924, *p* = 0.018). Kaplan–Meier survival curves stratified by primary tumor, PCI subgroups, and HIPEC use are shown in Figure 3, Figure 4, and Figure 5, respectively.

Multivariate Cox regression analysis identified primary tumor, PCI score, CC score, and HIPEC as independent predictors of overall survival (OS). Gastric cancer was associated with significantly worse survival compared to colorectal cancer (HR = 4.738, *p* < 0.001), whereas PMP correlated with improved survival (HR = 0.283, *p* < 0.001). A PCI score of 1–6 was linked to better survival outcomes (HR = 0.225, *p* < 0.001), while PCI scores of 7–12 had no significant effect. Incomplete cytoreduction (CC-2) was a strong negative prognostic factor (HR = 3.235, *p* < 0.001), whereas patients who received HIPEC demonstrated a survival benefit (HR = 0.649, *p* = 0.037). Detailed results are summarized in Table 3.

### 4.5. Disease-Free Survival (DFS)

The median DFS was 48 months. (Figure 6). Kaplan–Meier DFS curves stratified by primary tumor, PCI subgroups, and CC score are shown in Figure 7, Figure 8, and Figure 9, respectively.

In the multivariate Cox regression analysis, age remained a significant predictor, with each additional decade associated with a 17.5% reduction in the hazard of disease recurrence or death (HR = 0.825, 95% CI: 0.737–0.951, *p* = 0.010). A PCI of 1–6 was linked to a significantly lower hazard compared to PCI >19 (HR = 0.302, 95% CI: 0.177–0.514, *p* < 0.001). Incomplete cytoreduction (CC-2) was a strong negative prognostic factor, with a significantly higher hazard of disease recurrence or death compared to complete cytoreduction (CC0-1) (HR = 2.599, 95% CI: 1.532–4.410, *p* < 0.001). Detailed results are summarized in Table 4.

## 5. Discussion

Cytoreductive surgery (CRS), with or without hyperthermic intraperitoneal chemotherapy (HIPEC), has significantly improved survival outcomes for patients with peritoneal surface malignancies (PSM), particularly in high-income settings. However, its implementation in low- and middle-income countries (LMICs) has been limited by infrastructure constraints, lack of trained personnel, and financial limitations.

This study, the first multicenter analysis from North Africa, demonstrates that CRS and HIPEC are feasible and effective in a resource-limited setting, achieving 88% complete cytoreduction (CC-0/1)—a rate comparable to outcomes reported in high-income countries (HICs) [13,14,15]. Such findings emphasize the growing expertise in North African centers and provide a foundation for scaling up PSM programs in LMICs.

### 5.1. Impact of HIPEC on Disease-Free Survival (DFS) and Overall Survival (OS) by Tumor Origin

HIPEC significantly improved overall survival (OS), particularly when combined with complete cytoreduction (CC-0/1), with a median OS of 70 months for CRS+HIPEC vs. 64 months for CRS alone (*p* = 0.016). However, it did not significantly impact DFS, suggesting that while HIPEC extends survival, it does not consistently prevent recurrence.

DFS outcomes varied by tumor type. Pseudomyxoma peritoneii (PMP) and colorectal cancer (CRC), both chemosensitive, showed better DFS with HIPEC, whereas gastric cancer (GC) and mesothelioma, which are less responsive to chemotherapy, had shorter DFS despite HIPEC. Ovarian cancer (OC) showed heterogeneous DFS results, likely influenced by tumor burden and prior systemic therapy. These findings show the tumor-dependent effect of HIPEC, highlighting the importance of careful patient selection and tailored chemotherapy regimens.

### 5.2. Other Prognostic Factors for DFS

DFS was strongly influenced by tumor burden, with patients having PCI ≤ 6 experiencing significantly longer DFS than those with PCI > 19 (HR = 0.211, *p* < 0.001). Similarly, completeness of cytoreduction (CC score) was a major determinant, as CC-0/1 patients had significantly better DFS than CC-2 cases (HR = 2.599, *p* < 0.001), reinforcing that residual disease is a key predictor of recurrence.

In our study, older patients (>60 years) had a lower risk of recurrence compared to younger patients. This finding is consistent with Fugazzola et al. [16], who reported higher DFS in patients aged >60 undergoing CRS + HIPEC for gastric cancer (*p* = 0.016). This may be explained by more aggressive tumor biology in younger individuals, requiring further investigation to refine age-based treatment strategies. Many studies, however, did not find a significant association between age and DFS [17,18,19].

### 5.3. Predictors of Severe Postoperative Complications

Severe postoperative morbidity (Clavien–Dindo ≥ IIIa) occurred in 22% of patients, consistent with major HIPEC series. Independent predictors of morbidity included ASA score ≥ 2 (OR = 2.23, *p* = 0.003), spleno-pancreatectomy (OR = 3.80, *p* < 0.001), incomplete cytoreduction (CC-2) (OR = 3.83, *p* < 0.001), and HIPEC administration (OR = 2.36, *p* = 0.002).

Interestingly, CC-2 was linked to a higher complication rate than CC-0/1, contradicting the assumption that more extensive surgery leads to greater morbidity. This may be due to higher tumor burden, prolonged operative time, and increased surgical complexity in CC-2 cases. Additionally, technical limitations in achieving complete resection may lead to increased complications due to persistent disease infiltration. These findings emphasize the need for rigorous patient selection, improved surgical planning, and ongoing training to minimize morbidity [20].

### 5.4. The Learning Curve and Training Models in LMICs

The successful implementation of CRS+HIPEC in LMICs is closely linked to the learning curve associated with these complex procedures. In high-income countries, achieving proficiency typically requires 140–220 cases per center [21,22,23]. However, in North African centers, the learning curve varied based on mentorship availability and institutional support.

In Morocco, proficiency was achieved after 20–30 cases, with remote mentorship compensating for the lack of on-site expertise. In Tunisia, where CRS+HIPEC has been established since 2003, proficiency was reached faster (~10 cases per surgeon) due to longstanding experience and international fellowships. In Algeria, stable outcomes were achieved after ~20 cases, benefiting from on-site mentorship and structured training programs.

These findings highlight the importance of mentorship, standardized protocols, and structured training in shortening the learning curve [22]. Expanding regional collaborations and knowledge-sharing initiatives will be key to accelerating this process and improving patient outcomes in resource-limited settings.

### 5.5. Recommendations for Expanding CRS+HIPEC Programs in LMICs

The sustainable expansion of CRS+HIPEC in LMICs requires a strategic approach to address challenges such as limited specialized training, resource constraints, and inconsistent patient selection [24]. This can be achieved through three key pillars: training and mentorship, optimized HIPEC protocols, and multidisciplinary decision-making.

### 5.6. Standardized Training and Mentorship

Developing specialized training programs and mentorship initiatives is crucial for safe and effective CRS+HIPEC implementation [2]. International fellowships at high-volume HIPEC centers offer LMIC surgeons valuable exposure to standardized techniques [25]. Regional training hubs further enhance skill transfer by allowing surgeons to train in an environment similar to their own [26].

In addition to on-site programs, remote mentorship via telemedicine offers a scalable solution for early-phase centers. Virtual coaching, case discussions, and intraoperative guidance enable LMIC centers to benefit from expert oversight without the financial and logistical challenges of international fellowships. Hybrid training models, combining in-person and remote mentorship, can significantly reduce the learning curve and improve surgical safety.

### 5.7. Optimizing HIPEC Protocols for Resource-Limited Settings

Given the high costs of chemotherapy agents and perfusion equipment, LMICs must adapt HIPEC protocols to balance cost and efficacy. Switching from short-duration oxaliplatin to longer mitomycin-based perfusions has been shown to provide comparable oncologic benefits while reducing costs. Modifications in perfusion temperature, duration, and intraoperative chemotherapy selection should be tailored to regional patient profiles and available resources.

Standardizing perioperative care pathways is also essential to minimize morbidity and optimize recovery. Preoperative strategies—including nutritional support, thromboprophylaxis, and infection prevention—can reduce surgical stress and postoperative complications, which is especially critical in LMICs where prolonged hospital stays burden healthcare systems. Early recovery after surgery (ERAS) protocols should also be implemented to improve outcomes and optimize resource utilization [27,28].

### 5.8. Strengthening Multidisciplinary Decision-Making

Effective patient selection is key to optimizing CRS+HIPEC outcomes, particularly in LMICs with resource limitations. Multidisciplinary tumor boards (MDTs) ensure that only patients who will truly benefit undergo CRS+HIPEC, reducing futile procedures. Structured MDT meetings involving surgical oncologists, medical oncologists, radiologists, and anesthesiologists improve evidence-based decision-making and standardization of eligibility criteria [29].

Beyond patient selection, prehabilitation programs play a crucial role in optimizing patients before surgery. By enhancing nutritional status, physical conditioning, and respiratory function, prehabilitation has been shown to reduce postoperative complications and hospital stays [20]. However, its adoption in LMICs remains limited, emphasizing the need for structured perioperative protocols.

A comprehensive framework for LMICs, as proposed by Souadka et al. [30], highlights standardized patient care pathways, interdisciplinary collaboration, and quality management as key components for successful CRS+HIPEC programs. Their model integrates preoperative assessment, complication management, and postoperative surveillance to improve patient outcomes. Additionally, quality management standards (QMS)—including safety checklists, digital patient monitoring, and structured postoperative follow-up—enhance procedural safety. Resource allocation, financial planning, and continuous evaluation are also critical for ensuring program sustainability.

By adopting these structured strategies, LMIC centers can enhance multidisciplinary coordination, optimize perioperative care, and improve long-term survival for CRS+HIPEC patients while overcoming financial and infrastructural constraints.

### 5.9. Study Strengths and Limitations

This study provides the first multicenter analysis of CRS and HIPEC outcomes in North Africa, offering real-world insights into the feasibility and oncologic efficacy of these procedures in a resource-limited setting. By including data from multiple institutions across Morocco, Tunisia, and Algeria, the study enhances generalizability and reflects the evolving expertise in LMICs. Additionally, the large sample size and use of standardized perioperative protocols strengthen the validity of our findings, making them relevant for other LMIC settings considering CRS+HIPEC implementation.

However, certain limitations must be acknowledged. First, the retrospective design is inherently subject to selection bias and potential variability in data collection across centers. Differences in HIPEC protocols, patient selection criteria, and access to adjuvant therapy may have influenced survival outcomes. Additionally, the lack of long-term quality-of-life (QoL) data limits our ability to assess functional recovery post-treatment. Lastly, while this study highlights the learning curve associated with CRS+HIPEC, future prospective research should evaluate how mentorship programs and standardized training pathways impact long-term surgical proficiency and patient outcomes.

Despite these limitations, this study provides strong evidence that CRS+HIPEC is feasible in LMICs, achieving survival outcomes comparable to high-income settings. The findings emphasize the importance of structured training, multidisciplinary collaboration, and resource optimization, serving as a benchmark for future studies aiming to expand CRS+HIPEC programs in similar resource-constrained environments.

## 6. Conclusions

This study provides strong evidence that CRS and HIPEC are feasible and effective in LMICs, achieving survival outcomes comparable to high-income settings. Despite challenges in infrastructure and expertise, North African centers demonstrated high rates of complete cytoreduction, underscoring the growing regional capacity for advanced peritoneal oncology.

While HIPEC significantly improved overall survival, its impact on DFS varied by tumor type, highlighting the need for careful patient selection and tailored chemotherapy strategies. The study also confirms that tumor burden, cytoreduction completeness, and surgical expertise are key determinants of outcomes.

Scaling CRS+HIPEC in LMICs requires structured training programs, adapted HIPEC protocols, and stronger multidisciplinary collaboration. Implementing standardized pathways and resource-optimized protocols will be essential for ensuring sustainability. Future research should focus on long-term quality of life, cost-effectiveness, and refining patient selection to further improve outcomes in resource-limited settings.

## Figures and Tables

**Figure 1 cancers-17-02113-f001:**
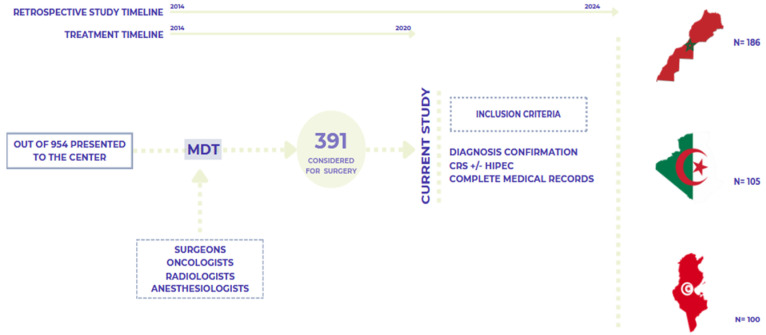
Study Design.

**Figure 2 cancers-17-02113-f002:**
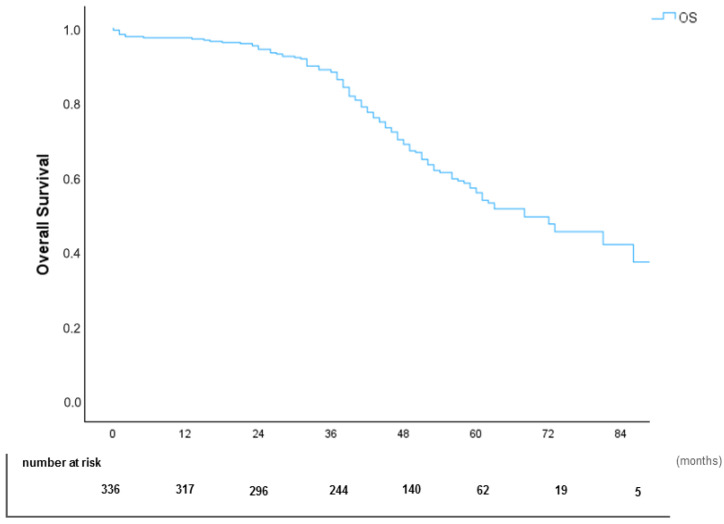
Kaplan–Meier Overall Survival (OS) Curve.

**Figure 3 cancers-17-02113-f003:**
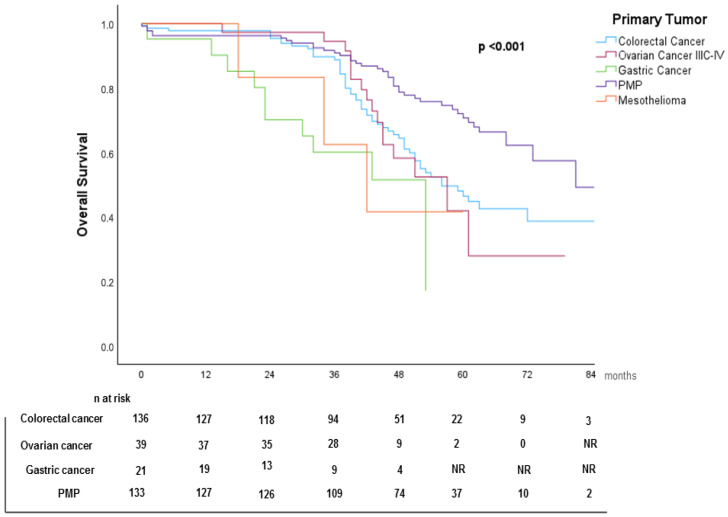
Kaplan–Meier overall survival (OS) curve stratified by primary tumor.

**Figure 4 cancers-17-02113-f004:**
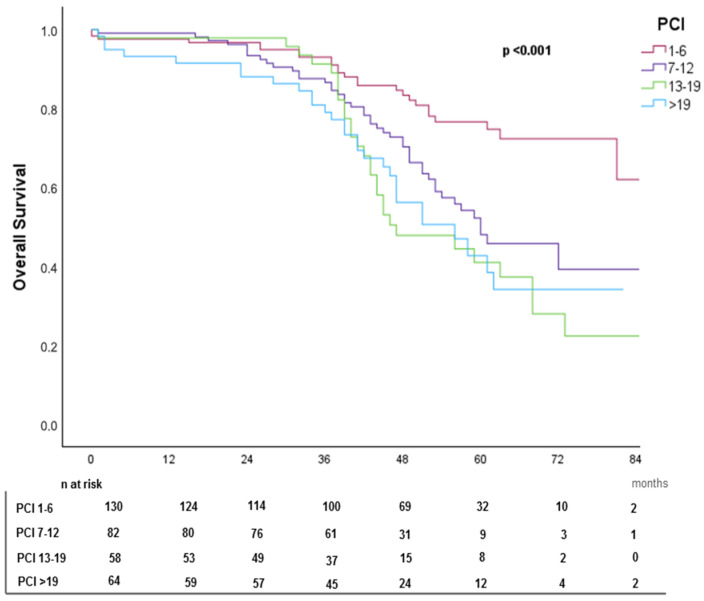
Kaplan–Meier overall survival (OS) curve stratified by PCI.

**Figure 5 cancers-17-02113-f005:**
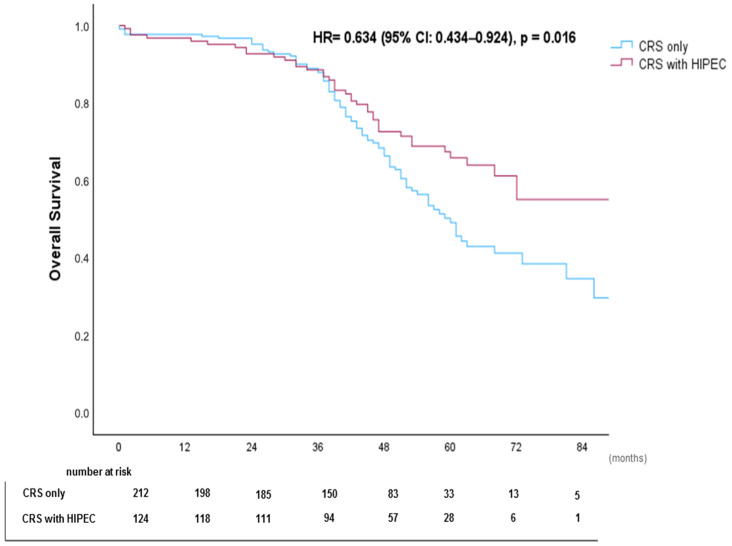
Kaplan–Meier Overall Survival (OS) Curve Stratified by HIPEC use.

**Figure 6 cancers-17-02113-f006:**
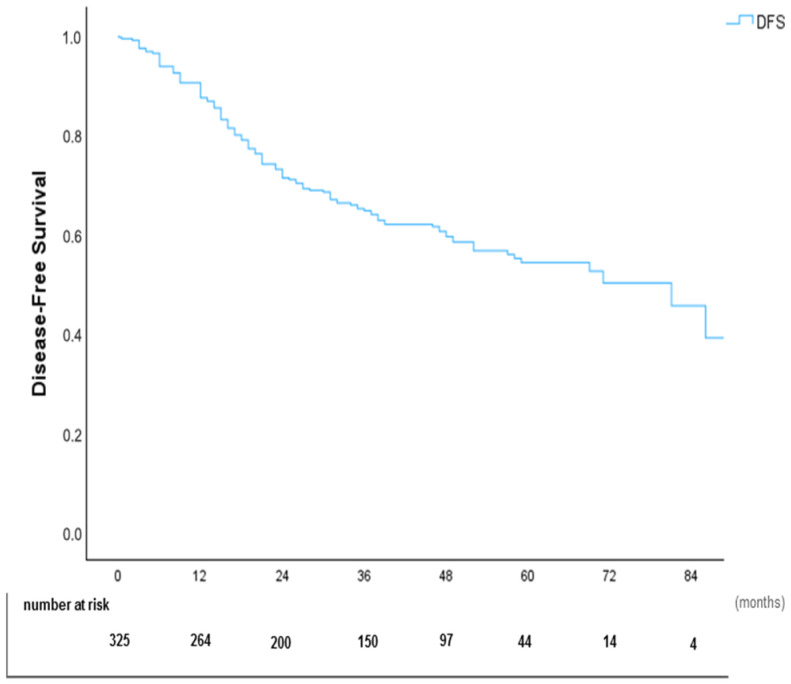
Kaplan–Meier disease-free survival (DFS) curve.

**Figure 7 cancers-17-02113-f007:**
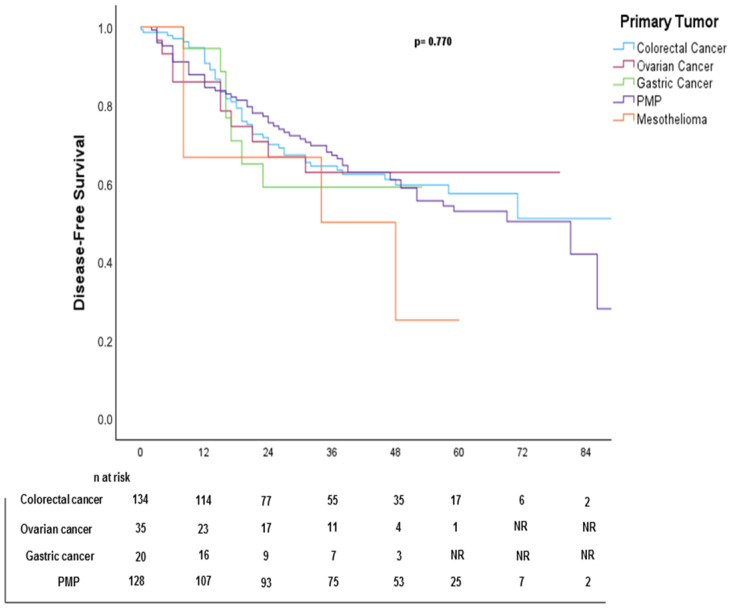
Kaplan–Meier disease-free survival (DFS) curve stratified by primary tumor.

**Figure 8 cancers-17-02113-f008:**
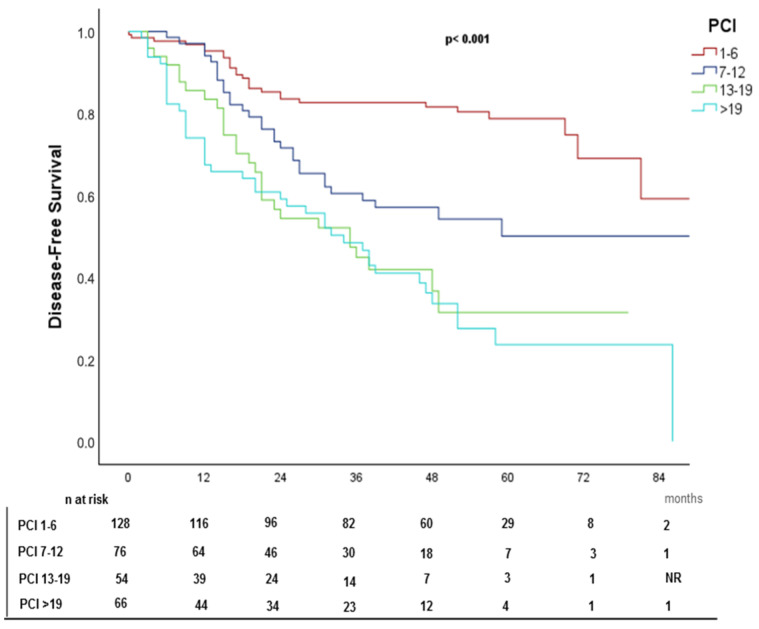
Kaplan–Meier disease-free survival (DFS) curve stratified by PCI.

**Figure 9 cancers-17-02113-f009:**
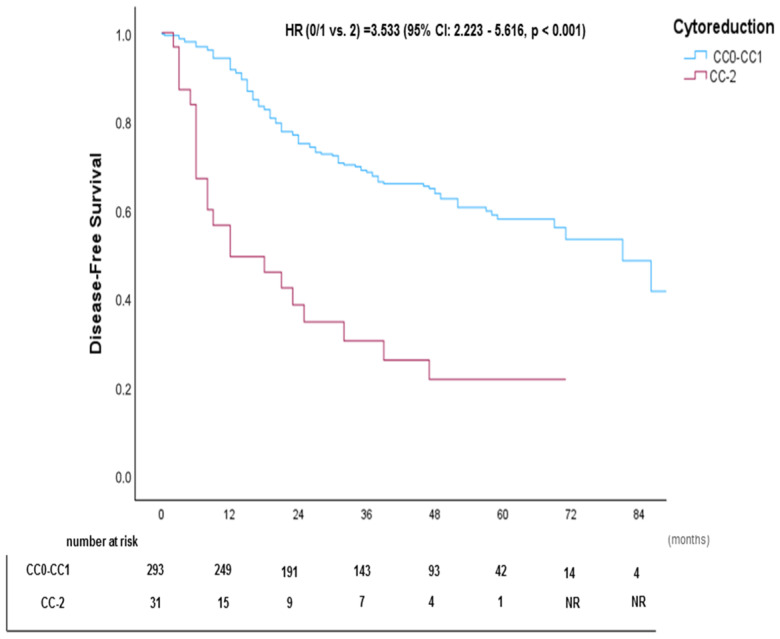
Kaplan–Meier disease-free survival (DFS) curve stratified by CC score.

**Table 1 cancers-17-02113-t001:** Overview of study population.

	Total (n = 391)	Morocco (n = 186)	Algeria (n = 105)	Tunisia (n = 100)	*p*-Value
**Age**	53 (44–64)	55 (45–65)	55 (46–62)	47 (39–57)	0.074
**Sex**					
Female	258 (66%)	125 (67.2%)	75 (71.4%)	58 (58%)	0.114
Male	133 (34%)	61 (32.8%)	30 (28.6%)	42 (42%)	
**ASA score**					
1	253 (64.7%)	136 (73.1%)	45 (42.9%)	72 (72%)	<0.001
≥2	138 (35.3%)	60 (26.9%)	60 (57.1%)	25 (28%)	
**ECOG status**					
0–1	385 (98.5%)	184 (98.9%)	101 (96.2%)	100 (100%)	0.067
≥2	6 (1.5%)	2 (1.1%)	4 (3.8%)	0 (0%)	
**Primary Tumor**					
CRC	168 (43%)	108 (58%)	28 (26.7%)	32 (32%)	<0.001
GC	25 (6.4%)	10 (5.4%)	3 (2.9%)	12 (12%)	
Ovarian	47 (12%)	32 (17.2%)	15 (14.3%)	0 (0%)	
PMP	144 (36.8%)	36 (19.4%)	56 (53%)	52 (52%)	
Mesothelioma	7 (1.8%)	0 (0%)	3 (2.9%)	4 (4%)	
**PCI**					
1–6	147 (37.6%)	72 (38.7%)	31 (30%)	44 (44%)	<0.001
7–12	99 (25.3%)	54 (29%)	17 (16.2%)	28 (28%)	
13–19	67 (17.1%)	38 (20.4%)	16 (15.2%)	13 (13%)	
>19	78 (19.9%)	22 (11.8%)	41 (39%)	15 (15%)	
**Procedure type**					
>4regions peritonectomy	231 (59.1%)	125 (67.2%)	70 (66.7%)	36 (36%)	<0.001
Proctectomy	108 (27.7%)	65 (35.1%)	28 (26.7%)	15 (15%)	0.001
Gastrectomy	40 (10.2%)	15 (10.5%)	11 (10.5%)	14 (14%)	0.29
Spleno-pancreatectomy	50 (12.8%)	11 (5.9%)	21 (20%)	18 (18%)	<0.001
Bowel Anastomoses >2	60 (15.3%)	32 (17.2%)	4 (3.8%)	24 (24%)	<0.001
Urologic procedures	21 (5.4%)	17 (9.1%)	0 (0%)	4 (4%)	0.003
**Cytoreduction**					
CC0-CC1	344 (88.2%)	170 (91.9%)	80 (20.5%)	94 (94%)	<0.001
CC2	46 (11.8%)	15 (8.1%)	25 (23.8%)	6 (6%)	
**Duration of Surgery (minutes)**	300 (210–360)	255 (180–300)	360 (325–420)	252 (170–330)	0.0768
**HIPEC**					
0	238 (60.9%)	167 (89.8%)	0 (0%)	71 (71%)	<0.001
1	153 (39.1%)	19 (10.2%)	105 (100%)	29 (29%)	
**Complication Status**					
0	226 (57.8%)	92 (49.5%)	66 (62.9%)	68 (68%)	0.005
1	165 (42.2%)	94 (50.5%)	39 (37.1%)	32 (32%)	
**Clavien-Dindo (III–V)**	87 (22.3%)	36 (19.4%)	39 (37.1%)	12 (12%)	<0.001
**Clavien-Dindo = V**	22 (5.6%)	12 (6.5%)	7 (6.6%)	3 (3%)	0.417
**Reoperation**	63 (16.8%)	27 (14.5%)	24 (22.9%)	12 (12%)	0.105

ASA: American Society of Anesthesiologists; ECOG: Eastern Cooperative Oncology Group. CRC: colorectal cancer; GC: gastric cancer; PMP: pseudomyxoma peritonei. PCI: Peritoneal Cancer Index; CC: Completion of Cytoreduction. HIPEC: hyperthermic intraperitoneal chemotherapy. *p*-values were calculated using Fisher’s Exact test for categorical variables and the Mann–Whitney U test for continuous variables.

**Table 2 cancers-17-02113-t002:** Predictive factors of severe complications identified by univariate and multivariate analysis.

	Severe Complications (Clavien–Dindo > III)		
Variable	Odds Ratio (OR)	95% Confidence Interval (CI)	*p*-Value
** *Univariate Analysis* **			
**Age ≥ 70**	1.275	0.612–2.654	0.517
**Sex (M vs. F)**	1.247	0.760–2.045	0.382
**ASA (≥2 vs. 1)**	2.750	1.688–4.480	**<0.001**
**ECOG status (≥2 vs. 0–1)**	7.277	1.310–40.423	0.023
**Primary Tumor**	-	-	0.340
CRC (Reference)	-	-	0.340
Gastric Cancer	1.342	0.496–3.631	0.562
Ovarian Cancer	0.872	0.372–2.045	0.752
PMP	1.635	0.496–3.631	0.562
Mesothelioma	0.708	0.082–6.092	0.753
**PCI**	-	-	**<0.001**
>19 (Reference)	-	-	<0.001
1–6	0.360	0.194–0.668	0.001
7–12	0.423	0.201–0.890	0.023
13–19	0.308	0.153–0.632	0.001
**Procedure**			
>4 regions peritonectomy	1.617	0.977–2.676	0.061
Protectomy	1.518	0.910–2.532	0.110
Gastrectomy	1.802	0.886–3.664	0.104
Spleno-pancreatectomy	4.500	2.423–8.356	**<0.001**
>2 Bowel Anastomoses	1.199	0.632–2.274	0.578
Urologic Procedures	0.353	0.081–1.546	0.167
**Cytoreduction (CC-2 vs. CC0-CC1)**	3.935	2.078–7.450	**<0.001**
**HIPEC (vs. No- HIPEC)**	3.180	1.943–5.205	**<0.001**
** *Multivariate Analysis* **			
**ASA (≥2 vs. 1)**	2.228	1.309–3.793	**0.003**
**Spleno-pancreatectomy**	3.803	1.765–8.197	**<0.001**
**Cytoreduction (CC-2 vs. CC0-CC1)**	3.830	1.765–8.297	**<0.001**
**HIPEC (vs. No HIPEC)**	2.364	1.369–4.080	**0.002**

ASA: American Society of Anesthesiologists; ECOG: Eastern Cooperative Oncology Group. CRC: colorectal cancer; GC: gastric cancer; PMP: pseudomyxoma peritonei. PCI: Peritoneal Cancer Index; CC: Completion of Cytoreduction. HIPEC: hyperthermic intraperitoneal chemotherapy. *p*-values for survival comparisons were calculated using the log-rank test. *p*-values represented in bold indicate statistical significance (*p* < 0.05).

**Table 3 cancers-17-02113-t003:** Prognostic factors of survival identified by univariate and multivariate analysis.

Variable	Hazard Ratio (HR)	95% Confidence Interval (CI)	*p*-Value
*Univariate Analysis*			
**Age**	0.990 (per 10-year increase)	0.860–1.138	0.880
**Sex (M vs. F)**	1.285	0.988–1.839	0.169
**ASA (≥2 vs. 1)**	1.328	0.935–1.888	0.113
**ECOG status (≥2 vs. 0–1)**	2.378	0.331–17.067	0.398
**Primary Tumor**	-	-	**<0.001**
CRC (Reference)	-	-	**<0.001**
Gastric Cancer	2.198	1.144–4.223	0.018
Ovarian Cancer	1.127	0.645–1.971	0.647
PMP	0.537	0.359–0.804	0.003
Mesothelioma	1.826	0.570–5.845	0.311
**PCI**	1.032 (for each 1-unit increase)	-	**<0.001**
>19	-	-	<0.001
1–6	0.406	0.247–0.666	<0.001
7–12	0.951	0.589–1.535	0.836
13–19	1.371	0.830–2.265,	0.217
**Cytoreduction (CC-2 vs. CC0-CC1)**	2.264	1.370–3.740	0.001
**HIPEC (vs. No- HIPEC)**	0.634	0.434–0.924	0.018
** *Multivariate Analysis: Independent Prognostic Factors of OS* **			
**Primary Tumor**	-	-	**<0.001**
CRC (Reference)	-	-	-
Gastric Cancer	4.738	2.334–9.618	**<0.001**
Ovarian Cancer	0.606	0.333–1.086	0.093
PMP	0.283	0.174–0.461	< 0.001
Mesothelioma	0.637	0.186–2.179	0.472
**PCI**	-	-	**<0.001**
>19 (Reference)	-	-	-
1–6	0.225	0.126–0.403	<0.001
7–12	0.671	0.397–1.135	0.137
13–19	1.638	0.933–2.874	0.086
**Cytoreduction (CC-2 vs. CC0-CC1)**	3.235	1.926–5.732	**<0.001**
**HIPEC (vs. No- HIPEC)**	0.649	0.432–0.975	**0.037**

ASA: American Society of Anesthesiologists; ECOG: Eastern Cooperative Oncology Group. CRC: colorectal cancer; GC: gastric cancer; PMP: pseudomyxoma peritonei. PCI: Peritoneal Cancer Index; CC: Completion of Cytoreduction. HIPEC: hyperthermic intraperitoneal chemotherapy. *p*-values for survival comparisons were calculated using the log-rank test. *p*-values represented in bold indicate statistical significance (*p* < 0.05).

**Table 4 cancers-17-02113-t004:** Prognostic impact of clinical factors on disease-free survival identified by univariate and multivariate analysis.

	Disease-Free Survival (DFS)		
Variable	Hazard Ratio (HR)	95% Confidence Interval (CI)	*p*-Value
** *Univariate Analysis* **			
**Age**	0.826 (per 10-year increase)	0.707–0.951	**0.005**
**Sex (M vs. F)**	1.319	0.924–1.881	0.127
**ASA (≥2 vs. 1)**	1.282	0.898–1.828	0.171
**ECOG status (≥2 vs. 0–1)**	3.776	0.924–15.433	0.064
**Primary Tumor**			
CRC (Reference)	-	-	0.784
Gastric Cancer	1.124	0.508–2.484	0.773
Ovarian Cancer	1.011	0.512–1.999	0.974
PMP	1.080	0.736–1.584	0.694
Mesothelioma	1.959	0.706–5.435	0.196
**PCI**	1.344 (for each 5-unit increase)	1.238–1.454	**<0.001**
>19	-	-	<0.001
1–6	0.211	0.130–0.344	<0.001
7–12	0.863	0.537–1.402	0.562
13–19	0.513	0.321–0.820	0.005
**Cytoreduction (CC-2 vs. CC0-CC1)**	3.533	2.223–5.616	<0.001
**HIPEC (vs. No- HIPEC)**	0.835	0.582–1.199	0.329
** *Multivariate Analysis: Independent Prognostic Factors of DFS* **			
**Age**	0.826 (per 10-year increase)	0.707–0.951	**0.010**
**PCI**	-	-	**<0.001**
>19 (Reference)	-	-	<0.001
1–6	0.302	0.177–0.514	<0.001
7–12	1.190	0.707–2.004	0.512
13–19	0.611	0.373–1.000	0.050
**Cytoreduction (CC-2 vs. CC0-CC1)**	2.599	1.532–4.410	**<0.001**

ASA: American Society of Anesthesiologists; ECOG: Eastern Cooperative Oncology Group. CRC: colorectal cancer; GC: gastric cancer; PMP: pseudomyxoma peritonei. PCI: Peritoneal Cancer Index; CC: Completion of Cytoreduction. HIPEC: hyperthermic intraperitoneal chemotherapy. *p*-values for survival comparisons were calculated using the log-rank test. *p*-values represented in bold indicate statistical significance (*p* < 0.05).

## Data Availability

The data presented in this study are available on request from the corresponding author. The data are not publicly available due to privacy restrictions.

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
