# Peer review of "Advancing Treatment Outcomes for Peritoneal Surface Malignancies in Low- and Middle-Income Countries: Insights from the First Multicenter Study in North Africa"

_cancers, 2025, doi:10.3390/cancers17132113_

Round 1

Reviewer 1 Report

Comments and Suggestions for Authors

Dear authors.

I have just single comment.

I would suggest to summarize the paper, mostly based on results section.

Try to think about the most representative findings of your research and select those tables / figures for the manuscript and pass the rest to supplementary material.

I am sorry, but the present version seems to be unnecessarily long.

Author Response

Response to Reviewer 1:

We thank the reviewer for the helpful comment and appreciate the suggestion regarding the length of the manuscript. As this is the first multicentric study of its kind in our region, we felt it was important to present a detailed account of our findings to provide the most value to readers and future researchers. Each result contributes to a better understanding of the complexity of peritoneal surface malignancies in low- and middle-income countries.

That said, we understand the importance of clarity and conciseness. If the editor or reviewer has specific suggestions regarding which tables or figures might be moved to supplementary material, we would be happy to consider them and make the necessary adjustments.

Reviewer 2 Report

Comments and Suggestions for Authors

From a biostats and clinical epidemiology point of view, here are some comments for the Authors

  • last pt has been enrolled in 2020, have you fixed a minimum follow-up to be inserted in the global multinational dataset?
  • continuous covariates have to be reported only as median/IQR
  • inferential analyses have to be redone by a non-parametric approach (i.e. Mann-Whitney and Fisher' exact tests)
  • standardized French treatment protocols, what do you mean? please, detail them carefully
  • any neoadjuvant regimens heve been administered? please, add this info
  • clavien-dindo, give all categories
  • colon and rectal cancers have to be reported as autonomous neoplasms
  • any anal cancer too? I suppose none, anyway add this info
  • logistic models: age and ASA can not be treated together! they are independent determinants
  • OS curves shows a clear time-dependent pattern, you have to prove that any determinant had not a similar behaviour
  • median follow-up for each cancer type is lacking
  • comparing different neoplasms families has a poor epidemiology sense and does not add any useful clinical info
  • the number of total and cancer-stratified OS and DFS events have to be added to results

Author Response

Response to Reviewer 2:

We would like to express our sincere gratitude to Reviewer 2 for the comprehensive and thoughtful evaluation of our manuscript. We have carefully considered all of the comments and have made the necessary revisions accordingly, as detailed below:

Comment:

last pt has been enrolled in 2020, have you fixed a minimum follow-up to be inserted in the global multinational dataset?

Response:

Yes, to ensure adequate follow-up for overall survival analysis, we excluded patients operated after December 2020. This allowed a minimum follow-up period to be established for all included patients. That said, all consecutive patients up to that cutoff date were enrolled in the study.

Comment:

continuous covariates have to be reported only as median/IQR

Response:

Thank you for your helpful comment. We have now corrected the presentation of continuous variables such as age and duration of surgery, and reported them as median (IQR) 

Comment:

"Standardized French treatment protocols, what do you mean? Please, detail them carefully."

Response:

Thank you for pointing this out. We apologize for the lack of clarity. By “standardized French treatment protocols,” we referred to perioperative and intraoperative approaches derived from established practices in high-volume French centers such as the Institut Gustave Roussy (IGR), with which our teams in Morocco and Tunisia have direct mentorship links. These include:

Patient selection criteria based on multidisciplinary tumor board (MDT) discussion using imaging, PCI scoring, and ECOG/ASA assessments.

CRS techniques based on Sugarbaker’s peritonectomy procedures and PCI-based resections.

HIPEC protocols including a short-duration oxaliplatin-based regimen (460 mg/m² over 30 minutes) for CRC/PMP/GC, and cisplatin-based perfusion for ovarian cancer.

Postoperative care aligned with ERAS principles, including early mobilization, thromboprophylaxis, infection control, and nutritional support.

We have now clarified this in the revised manuscript (Methods > Center-Specific Protocols and Practices section).

Comment :

"Any neoadjuvant regimens have been administered? Please, add this info."

Response:

We appreciate this important question. In our cohort, systemic neoadjuvant chemotherapy was selectively administered, depending on tumor type and institutional MDT recommendations:

For colorectal cancer, neoadjuvant FOLFOX or FOLFIRI was given in patients with high PCI or suspected nodal involvement.

For gastric cancer, perioperative chemotherapy (based on the ECX and FLOT regimen) was used.

For ovarian cancer, when needed patients received neoadjuvant carboplatin-paclitaxel chemotherapy prior to interval CRS.

For pseudomyxoma peritonei and mesothelioma, no systemic neoadjuvant therapy was administered.

We have now included this information in the revised Methods > Patient Selection section.

Comment:

clavien-dindo, give all categories

Response:

We thank the reviewer for this suggestion. However, given the retrospective nature of the study and to preserve clarity in the presentation of results, we chose to report only severe complications defined as Clavien-Dindo grade IIIb and above. We also presented grade V (mortality) separately from other complications for better clinical interpretation.

Comment: 

colon and rectal cancers have to be reported as autonomous neoplasms

Response:

We appreciate the reviewer’s remark. In line with similar studies on peritoneal surface malignancies, we grouped colon and rectal cancers under a single entity of colorectal adenocarcinoma, as they share similar histology, management principles, and patterns of peritoneal spread.

Comment:

logistic models: age and ASA can not be treated together! they are independent determinants

Response:

We fully agree with the reviewer’s comment. In our analysis, age ≥70 years was defined as the threshold for elderly patients and was treated as an independent variable, not associated with ASA status, since both are considered distinct clinical determinants.

Comment:

comparing different neoplasms families has a poor epidemiology sense and does not add any useful clinical info

Response:

Thank you for this important observation. We acknowledge the epidemiological heterogeneity between different primary tumors. However, in line with several foundational publications in the field (e.g., references 10 and 11), initial multicentric studies on peritoneal surface malignancies often begin by comparing different neoplasm families. This approach provides a global overview of clinical relevance in the studied context before conducting disease-specific analyses. It is for this reason that we adopted a similar structure in our work.

Comment

"The number of total and cancer-stratified OS and DFS events have to be added to results."

Response:

Thank you for this valuable suggestion. We have now added the total and stratified numbers of OS and DFS events in the Results section

Total OS events (deaths): 146 (37.3%)

OS events by tumor type:

CRC: 72

PMP: 34

OC: 22

GC: 16

Mesothelioma: 2

Total DFS events (recurrence or death): 173 (44.2%)

DFS events by tumor type:

CRC: 84

PMP: 39

OC: 26

GC: 21

Mesothelioma: 3

These data are now incorporated in the revised Results section.

Reviewer 3 Report

Comments and Suggestions for Authors

I was very pleased to read the study by respected Amine Souadka and co-authors. The article is devoted to the study of advancing treatment outcomes for peritoneal surface malignancies in low- and middle-income countries in north Africa. Peritoneal carcinomatosis is a threatening complication of ovarian cancer and gastrointestinal cancer, and hyperthermic intraperitoneal chemotherapy can significantly improve outcomes. HIPEC is beginning to be actively used in developed countries. However, its implementation in low- and middle-income countries has been limited by infrastructure constraints, lack of trained personnel, and financial limitations. This study, the first multicenter analysis from North Africa, demonstrates that HIPEC are feasible and effective in a resource-limited setting. Such findings emphasize the growing expertise in North African centers and provide a foundation for scaling up peritoneal surface malignancies programs. A great advantage of the work is the availability of data from several countries, which allows us to evaluate different experiences. Therefore, I consider the authors' study to be very relevant and interesting for readers.

The manuscript consists of standard sections (IMRAD), is well structured, presented in a very clear and logical manner. The article is easy to read and easy to perceive, which is its undoubted advantage. The manuscript cites 27 works, of which about 30% are from the last five years. Thus, the manuscript contains up-to-date information.

In their study, the authors processed data from a large number of patients (almost 400 cases). The study design, selection of patients, choice of data and methods of statistical processing seem adequate to me. The methods are described in detail and allow the experiments to be reproduced. The manuscript contains eight figures and four tables, which correctly reflect the main results of the article and are easy to interpret. The authors' conclusions are supported by the results. The discussion is written in an interesting and scientifically sound manner.

I highly appreciate this study and believe that it can be extended to other surgical interventions and other countries. I have several suggestions for improving the article that I encourage the authors to consider.

Major comments

  1. In the introduction, lines 59–66 require references to support the sentences. Please provide some up-to-date references that reflect the experience of other countries.
  2. In the Methods section, I would ask for a figure or scheme that reflects the design of the experiment.
  3. There are many abbreviations in the article. I would recommend making a list of abbreviations at the end of the article.

Minor comments

  1. In the footnotes to tables 1, 2, 3, 4, please provide all the abbreviations.
  2. For all tables, in the footnotes, indicate the criterion by which the p-value was calculated.
  3. In tables 2, 3, 4, some p-values are highlighted in bold. Does this have any significance? If so, indicate it in the footnotes.

Author Response

Response to Reviewer 3:

We sincerely thank Reviewer 3 for the thoughtful and encouraging feedback. We are particularly grateful for your kind words regarding the relevance, clarity, and contribution of our study to the field of peritoneal surface malignancies (PSM) in low- and middle-income countries. Your recognition of the challenges and importance of advancing HIPEC implementation in North Africa is deeply appreciated.

We have carefully considered all your suggestions and provide our detailed responses below:

Major Comments

  1. Introduction (lines 59–66) – Add references:
    Thank you for pointing this out. We have now added up-to-date references that reflect the experience of other countries using HIPEC and the global evolution of PSM programs. These additions help place our study within the broader international context.

  2. Methods – Add a figure/scheme for study design:
    As suggested, we have added a schematic figure outlining the study design, including patient selection, data collection, and analysis workflow. We believe this will enhance clarity for readers.

  3. List of abbreviations:
    We have now included a comprehensive list of abbreviations at the end of the manuscript to facilitate easier reading and comprehension.

Minor Comments

  1. Footnotes of Tables 1–4 – Define all abbreviations:
    We have revised the footnotes of all relevant tables to ensure that all abbreviations are clearly defined.

  2. Footnotes – Specify test used for p-values:
    We have indicated the specific statistical test used to calculate the p-values in the footnotes of each table for transparency and reproducibility.

Bold p-values in Tables 2–4 – Clarify meaning:
We have bolded the p values that were <0.05 (indicating statistical significance) just to make them relevant and easier to read from the table. We have now added a clear note in the footnotes of each relevant table to explain this formatting choice.

Round 2

Reviewer 1 Report

Comments and Suggestions for Authors

Dear authors.

In my view, figure 1 should be supplementary material, as well as table 1.

I would keep the other three tables, and would try to select some, but not all,. of the Kaplan-Meier curves.

Author Response

Dear Reviewer, thank you very much for taking time to analyse our manuscript. We received other comments from the two other reviewers asking to add some tables and add the figure one as the study design?

I honestly do agree that it may appear long, however in order to be complete and answer all the comments, we think that we may keep these both elements.

Again We thank you for all your t helpful comments.

Reviewer 2 Report

Comments and Suggestions for Authors

I have no more concerns

Author Response

Thank you very much for your surpport.

Reviewer 3 Report

Comments and Suggestions for Authors

The authors have done a good job and have taken my comments into account to the best of their ability. I am completely satisfied. The article may be accepted for publication.

Author Response

Thank you very much for your surpport.